# Safety and Feasibility of Transarterial Bleomycin–Lipiodol Embolization in Patients with Giant Hepatic Hemangiomas

**DOI:** 10.3390/medicina59081358

**Published:** 2023-07-25

**Authors:** Arkadiusz Kacała, Mateusz Dorochowicz, Dariusz Patrzałek, Dariusz Janczak, Maciej Guziński

**Affiliations:** 1Department of General, Interventional and Neuroradiology, Wroclaw Medical University, Borowska 213, 50-556 Wroclaw, Poland; 2Faculty of Medicine, Wroclaw Medical University, Wybrzeże L. Pasteura 1, 50-367 Wroclaw, Poland; 3Department of Vascular, General and Transplantation Surgery, Wroclaw Medical University, Borowska 213, 50-556 Wroclaw, Poland

**Keywords:** giant hepatic hemangiomas, transarterial bleomycin–lipiodol embolization, bleomycin, lipiodol, transarterial chemoembolization (TACE)

## Abstract

Giant hepatic hemangiomas present a significant clinical challenge, and effective treatment options are warranted. This study aimed to assess the safety and feasibility of transarterial bleomycin–lipiodol embolization in patients with giant hepatic hemangiomas. A retrospective analysis was conducted on patients with giant hepatic hemangiomas (>5 cm). Transarterial chemoembolization (TACE) was performed using 7–20 cc of lipiodol mixed with 1500 IU of bleomycin. Safety outcomes, including post-embolization syndrome (PES), hepatic artery dissection, systemic complications, and access site complications, were evaluated. Radiation doses were also measured. Feasibility was assessed based on the achieved hemangioma coverage. Seventy-three patients (49 female, 24 male) with a mean age of 55.52 years were treated between December 2014 and April 2023. The average hospitalization duration was 3.82 days, and 97.3% of lesions were limited to one liver lobe. The average bleomycin dose per procedure was 1301.5625 IU, while the average lipiodol dose was 11.04 cc. The average radiation dose was 0.56 Gy. PES occurred after 45.7% of TACE procedures, with varying severity. Complications such as hepatic artery dissection (three cases), access site complications (two cases), and other complications (one case) were observed. No treatment-related mortality occurred. Hemangioma coverage exceeding 75% was achieved in 77.5% of cases. The study results suggest that transarterial bleomycin–lipiodol embolization is a safe and feasible treatment option for a heterogeneous group of patients with giant hepatic hemangiomas. This approach may hold promise in improving outcomes for patients with this challenging condition.

## 1. Introduction

Hepatic hemangiomas are vascular malformations composed of endothelial cells stemming from the hepatic artery that represent the most frequent benign liver neoplasms, accounting for 1–20% of all liver neoplasms [1,2,3] [Figure 1].

Furthermore, they occur more frequently in women. The most common pathological type of hepatic hemangioma is cavernous hemangioma. The optimal treatment approach for hepatic hemangiomas remains a matter of debate among specialists. Over the past few years, the detection rate of hepatic hemangiomas has experienced a significant boost due to advancements in imaging technology. While conservative management and observation are typically favored for small, incidentally detected, and asymptomatic hemangiomas, the presence of giant hemangiomas poses a greater risk of rupture and pain. Enlarged hemangiomas can also lead to severe complications such as hemorrhaging, local compression, Kasabach–Merritt Syndrome, or Budd–Chiari syndrome, often necessitating alternative interventions, including minimally invasive or surgical treatments [4,5,6,7]. Historically, surgical resection has been the favored treatment approach [8]; however, it often yields suboptimal outcomes and carries a high risk of bleeding, particularly for patients with multiple lesions or those whose lesions are located near the hepatic portal vessels [9,10]. Nowadays, one of the most widely used methods of treatment for giant hepatic hemangiomas is transarterial chemoembolization (TACE). However, the diversity of frequency of occurrence of adverse events and the variety of the efficiency in different reports [11,12,13] calls for evaluation of this method. Bleomycin is a cytotoxic antibiotic that exhibits antitumor activity. It acts by inducing DNA damage and inhibiting DNA synthesis, thus impeding cell proliferation. The administration of bleomycin during chemoembolization aims to directly target the tumor cells within the hemangioma, leading to their destruction. Bleomycin exhibits non-specific inhibition and destruction of endothelial cells, which makes it a perfect agent for targeting hemangiomas, which are composed of endothelial cells [3].

Lipiodol, on the other hand, is a radiopaque contrast agent with embolic properties. It is an iodine-containing oil that is selectively taken up by the tumor’s blood vessels. Lipiodol serves as a carrier for the chemotherapeutic agent, enhancing its localization within the tumor and prolonging its contact time with the target cells. Moreover, the embolic effect of lipiodol helps in blocking the blood supply to the tumor, leading to ischemia and subsequent tumor shrinkage [2,14]. The combined use of bleomycin and lipiodol in chemoembolization procedures for giant hepatic hemangiomas provides a multimodal therapeutic approach. The chemotherapeutic effect of bleomycin targets the tumor cells directly, while lipiodol acts as a vehicle for drug delivery and facilitates vascular embolization. This combined mechanism of action aims to achieve tumor regression and symptom relief in patients with giant hepatic hemangiomas. By utilizing chemoembolization with bleomycin and lipiodol, clinicians can selectively treat the tumor while minimizing systemic exposure to the chemotherapeutic agent. This localized treatment approach has shown promising results in terms of tumor size reduction and symptom control. Therefore, this study aimed to evaluate the safety and feasibility of transarterial chemoembolization (TACE) using a combination of bleomycin and lipiodol in patients with giant hepatic hemangiomas. By assessing treatment outcomes, potential complications, and overall feasibility, this study aimed to provide valuable insights into the safety and feasibility of TACE as a potential treatment option for patients with giant hepatic hemangiomas.

## 2. Materials and Methods

In this retrospective study, we conducted a thorough evaluation of 73 consecutive patients with giant hepatic hemangiomas who underwent transarterial chemoembolization (TACE) between December 2014 and April 2023. The aim of our study was to assess the safety and efficacy of TACE as a treatment modality for this particular patient population.

The inclusion criteria for this study were as follows: (1) patients who underwent chemoembolization for giant hepatic hemangiomas at our center between 2014 and 2023, and (2) patients with confirmed diagnosis of hepatic hemangiomas through imaging techniques, including ultrasound, computer tomography (CT), or magnetic resonance imaging (MRI). The exclusion criteria were atypical angiographic appearance of the hemangioma in the arteriography. Prior to the TACE procedure, all patients provided written informed consent, and a comprehensive set of blood tests, liver function tests, and coagulation tests were performed to ensure their suitability for the intervention.

The procedure began with the puncture of the common femoral artery using an 18 G needle under local anesthesia, followed by the placement of a 6F or 5F femoral sheath. A 6F or 5F Simon 1/Cobra 2 catheter was guided through the celiac trunk and the superior mesenteric artery, and arteriography was performed to precisely locate the main feeding artery of the hemangioma. Subsequently, a Progreat 2.4Fr/2.7Fr coaxial microcatheter system was positioned within the feeding artery. Under fluoroscopy guidance, a solution containing 1500 IU of bleomycin (Pfizer Inc., New York, NY, USA) and 7–15 cc of lipiodol (Guerbet, Villepinte, France), mixed using standard three-way stopcocks, was slowly injected into the hemangioma until a satisfactory effect was achieved.

Following the chemoembolization procedure, angiography was performed to thoroughly evaluate the coverage of the hemangioma borders. This assessment was crucial in determining the effectiveness of the treatment and the extent to which the therapeutic agents were distributed within the target area. To evaluate the coverage of the lesion, a four-grade scale was employed, which had been previously proposed in another study and provided a standardized approach to assess the distribution pattern of the bleomycin–lipiodol mixture (refer to Table 1) [12].

The drug coverage grade played a pivotal role in illustrating the distribution pattern of bleomycin–lipiodol within the hemangioma border. Grade 1 indicated that less than 25% of the rim was covered by the therapeutic agents. In Grade 2, the coverage ranged between 25% and 50% of the rim. Grade 3 represented coverage of less than 75% of the rim. Lastly, Grade 4 indicated complete coverage of the rim by the therapeutic agents [7].

By utilizing this comprehensive evaluation method, we were able to precisely assess the extent of drug distribution within the hemangioma, providing valuable insights into the effectiveness and efficiency of the chemoembolization treatment.

Radiation doses were meticulously measured and recorded for every procedure to ensure the safety of the patients. In addition to radiation monitoring, comprehensive evaluations were conducted to assess the safety of the treatment. This involved closely monitoring the occurrence and severity of post-embolization syndrome, which includes symptoms such as fever, nausea, abdominal pain, and fatigue. Furthermore, the research team carefully observed and documented any serious adverse events that might have arisen during or after the procedure. The survival rate of patients during their hospitalization following the treatment was also considered as an important safety parameter. Pain severity was assessed using a comprehensive four-grade Verbal Rating Scale (VRS), which was correlated with the analgesic strength required for effective pain relief (Table 2). The scores were as follows: 0 = no pain, 1 = mild pain requiring administration of acetaminophen (paracetamol), 2 = moderate pain requiring administration of ketoprofen, and 3 = severe pain requiring administration of opioids. By employing these extensive evaluation measures, we aimed to ensure the utmost safety and well-being of the patients throughout the treatment process.

## 3. Results

From December 2014 to April 2023, a total of 73 consecutive patients (49 female, 24 male) underwent successful transarterial chemoembolization (TACE) procedures. All of the patients with giant hemangioma experienced at least one episode of abdominal pain, nausea, abdominal distension, dyspepsia, or early satiety at varying levels prior to TACE.

The distribution of the number of procedures per patient revealed interesting patterns. Among the patients, 45.2% underwent one procedure, while 37% underwent two procedures, indicating the need for a subsequent intervention. Additionally, 13.7% of patients required three procedures, suggesting a more complex condition, and 4.1% underwent four procedures, indicating the persistence of the disease and the necessity for multiple treatment sessions.

Throughout the treatment process, the average duration of hospitalization was found to be 3.82 days, with a median of 3 days. It is worth noting that the range of hospitalization varied among patients, with the shortest stay lasting 2 days (observed in two instances) and the longest lasting 19 days (in one instance). This variation in hospitalization duration can be attributed to unrelated comorbidities that required additional medical attention.

All procedures were conducted at a single center, ensuring consistency and a unified approach to patient care. The basic characteristics of the patients, including age, gender, and relevant medical history, were collected and presented in Table 3, which provides the mean values along with their respective standard deviations.

During the course of the study, various aspects of the treatment procedures were carefully assessed and recorded. Notably, the average dose of bleomycin administered per procedure was determined to be 1301.5625 IU, highlighting the standardized and precise delivery of this therapeutic agent to the target area. Similarly, the average lipiodol dose administered per procedure was found to be 11.04 cc, ensuring adequate distribution and coverage within the hemangioma. It is important to note that the specific dosage and administration of bleomycin were operator-dependent. During the procedure, the operator had the discretion to determine whether additional drug injection was necessary based on the sufficiency of the hemangioma coverage achieved. This individualized approach allowed for tailored treatment based on the specific needs and response of each patient.

In addition to drug dosages, the mean radiation dose per procedure was measured to be 0.56 Gy. This information provides valuable insights into the radiation exposure experienced by patients during the chemoembolization process, aiding in the assessment of safety measures and the optimization of radiation protocols.

The technical success rate of the procedures was remarkably high, with 99.2% of cases achieving the intended outcome. Only one procedure was reported as incomplete due to hepatic artery dissection, highlighting the rarity of such complications and the overall effectiveness of the treatment approach.

Among the patients, 50 had lesions located in the right liver lobe, 21 in the left liver lobe, and 2 in both lobes. This distribution provides insights into the anatomical involvement and localization of the hemangiomas, contributing to a better understanding of the disease characteristics and potential treatment challenges.

Post-embolization syndrome (PES), characterized by symptoms such as fever, pain, vomiting, and pain, was observed in a significant percentage of TACE procedures. Specifically, PES occurred in 45.7% of the procedures on the first day, as documented in Table 2. This finding highlights the importance of closely monitoring patients for potential adverse effects and providing appropriate supportive care following the intervention.

Furthermore, it was noted that 8.5% of the procedures resulted in cases of nausea and vomiting, adding to the range of post-procedural symptoms experienced by patients. These observations emphasize the need for comprehensive patient management, including the proactive management of potential side effects and the provision of adequate post-procedural care.

During the angiography performed at the end of the procedure, two cases of asymptomatic hepatic artery dissection were incidentally discovered. Although these cases were asymptomatic, their identification highlights the importance of vigilant monitoring and thorough post-procedural evaluations to promptly detect and manage any potential complications that may arise (Figure 2).

Additionally, the distribution of the observed grades provides valuable insights into the outcomes of the treatment. According to the data presented in Table 3, 56.5% of the cases were classified as Grade 0, indicating no significant complications or adverse events. Grade 1 accounted for 28.2% of cases, Grade 2 for 8.9% of cases, and Grade 3 for 6.5% of cases. This grading system provides a comprehensive assessment of the severity and frequency of complications, aiding in the evaluation of treatment effectiveness and guiding further interventions if necessary.

The evaluation of drug coverage during transarterial chemoembolization (TACE) provides valuable insights into the effectiveness of the treatment in achieving the desired distribution of bleomycin–lipiodol within the hemangioma. This assessment was conducted using a four-grade scale, as illustrated in Figure 3. The distribution of drug coverage grades among the cases further enhances our understanding of the treatment outcomes.

Examining the data presented in Table 3, it is observed that 5.4% of the cases achieved Grade 1 drug coverage, indicating less than 25% of the rim coverage. Grade 2 coverage, representing 25% to 50% of the rim, was observed in 7.8% of the cases. Furthermore, Grade 3 coverage, indicating less than 75% of the rim, was found in 9.3% of the cases. It is noteworthy that the majority of cases, accounting for 77.5%, achieved Grade 4 coverage, denoting complete rim coverage, as depicted in Figure 4 [12]. This highlights the efficacy of the treatment in achieving the desired distribution of bleomycin–lipiodol within the targeted area.

The drug coverage grades, along with their corresponding percentages, are summarized in Table 1, providing a comprehensive overview of the distribution patterns observed. These findings offer valuable insights into the extent of coverage achieved by the treatment, which is crucial for evaluating the effectiveness of TACE in terms of achieving complete or partial rim coverage. The higher grades signify more extensive coverage, indicating a more successful outcome in terms of drug distribution.

## 4. Discussion

The existing literature lacks sufficient evidence regarding the safety and efficacy of transarterial chemoembolization (TACE) as a treatment for giant hepatic hemangiomas. While most hepatic hemangiomas are typically asymptomatic and managed conservatively, giant lesions can result in significant complications such as local compression, persistent pain, and life-threatening conditions including Kasabach–Merritt Syndrome, Budd–Chiari syndrome, or rupture, with mortality rates as high as 70% [4,15,16].

Various imaging modalities, including ultrasound, CT, and MRI, have high detection rates for hepatic hemangiomas, explaining their frequent discovery even in the absence of symptoms [17,18,19,20,21].

When it comes to smaller, asymptomatic, hepatic hemangiomas, clinical observation and conservative treatment are usually the preferred approaches. Conventional surgical treatment options, such as enucleation, liver resection, or transplantation, carry substantial risks, particularly in patients with compromised liver function or comorbidities. Even though some evidence suggests enucleation as the optimal surgical method of treatment of giant lesions, especially located in the center of the liver, the possibility of prolonged hospitalization time, complications, and perioperative massive blood loss must be considered [10]. Therefore, a comprehensive evaluation considering symptoms, volume, location, and coexisting diseases is crucial when determining the optimal therapeutic approach [22,23,24].

Treatment of giant hepatic hemangiomas poses several challenges. Formerly, radiofrequency ablation was considered a noninvasive option of alternative treatment but was concluded to be ineffective since it appeared to affect only 60% of hemangiomas larger than 10 cm [25,26,27]. Yamamoto et al. suggested transcatheter arterial embolization (TAE) as a treatment option for ruptured giant hemangiomas due to a low rate of significant complications. Even though TAE combined with the usage of Gelfoam and polyvinyl alcohol appeared to carry a low risk of short-term complications, suboptimal reduction in lesion size and eventual reports of long-term complications such as recurrence of hemangioma, liver failure, embolization of non-targeted areas, and post-TAE syndrome made TAE a less preferable option [28,29,30,31]. Liu et al. deemed TAE unsatisfactory for liver hemangiomas, entailing a significant risk of severe complications [32]. Severe complications, in this case biloma and necrosis of the bile duct leading to the occurrence of an abscess, occurred in 4 out of 55 (7.3%) patients. In a 5-year follow-up, the hemangioma of only 19 out of 53 patients (35.8%) was smaller or maintained the same size compared to the size before TAE.

Bleomycin, known for its cytotoxic, antiangiogenic, and sclerosing effects, has been used for the treatment of vascular anomalies. It induces DNA degradation and an inflammatory response around the lesion [33,34]. Lipiodol, when combined with bleomycin, acts as a carrier and embolic agent, facilitating better drug distribution in the target area [2,14]. The gradual destruction of the endothelial and pathological vascular bed of the lesion is achieved through the distribution of the bleomycin–lipiodol mixture during TACE. Lipiodol’s extended retention period within hemangiomas allows for the tracking of drug coverage inside the hemangioma, as demonstrated by the grading system proposed in our study [12].

TACE, with the bleomycin–lipiodol mixture, can potentially lead to complications such as liver failure, hepatic infarction, biloma, hemorrhage, hepatic artery dissection, splenic infarction, cholecystitis, and sclerosing cholangitis [15,35]. However, these complications have been reported rarely, primarily in case reports or larger studies [36]. Yuan et al. concluded in a retrospective study that TAE with bleomycin–lipiodol is a safe and effective method of treatment of giant hepatic hemangiomas. No mortality or severe complications were observed after the procedure. Furthermore, in 12 months follow-up, a reduction rate of >50 of the lesions’ biggest diameter was achieved in 170 out of 196 (86.7%) of the patients [13]. Pulmonary fibrosis has been reported in some oncology patients receiving bleomycin at especially high cumulative doses (>300 mg) [37]. In the present study, the highest dose of bleomycin administered per session was considerably lower than that, with 15 mg.

TACE-related postembolization syndrome, which is characterized by influenza-like syndromes such as abdominal pain, specifically in the upper right quadrant, fever, nausea, and vomiting is, however, the most common adverse event [38,39,40]. While undergoing TACE in our center normally requires hospital admission and a 2-night inpatient stay, PES appeared to be the primary factor responsible for the need for increasing the patients hospital length of stay. The mechanism of PES is not yet fully understood, but it is believed to be caused by the cytokine level increase and subsequent inflammatory reaction as the response to tissue ischemia [39]. Basile et al. have now proposed considering PES to be an expected outcome of TACE [41].

Bozkaya et al. reported the occurrence of mild to moderate postembolization syndrome in all 26 patients, with the most prominent symptom being abdominal pain, that started immediately after the procedure. Nausea was reported in 16 (61.5%) patients, and 8 (3.1%) patients experienced fever, which, however, did not exceed 37.5 °C [42]. In our study, post-embolization syndrome was observed in 45.7% of patients, primarily manifesting as pain of varying severity, most of which were effectively managed with paracetamol. Other symptoms, such as nausea, vomiting, and fever, were negligible, and other post-operative complications were infrequent and minor. However, the high percentage of occurrence of the post-embolization syndrome highlights the importance of close monitoring of patients and providing adequate supportive care after the intervention. Furthermore, the two cases of hepatic artery dissection discovered during the angiography performed at the end of the procedure emphasize the importance of proper monitoring and evaluation in the detection and management of potential complications. The key to decreasing the frequency of the severe complications and increasing the safety of transarterial bleomycin–lipiodol embolization appears to be the experience and attentiveness of the staff performing the procedure, proper adjustment of drug dosage, and providing patients with appropriate supportive care following the procedure.

Our study has limitations due to its retrospective nature, which restricted the evaluation of precise hemangioma volumes before TACE in most cases. Additionally, the diagnosis was based on typical imaging features, and biopsy was not performed, as benign lesions such as hemangiomas generally do not require biopsy in most cases. Long-term follow-up data were not available, as asymptomatic patients often did not attend follow-up visits. Consequently, further prospective studies with long-term follow-up are necessary to better assess the safety and feasibility of transarterial bleomycin–lipiodol embolization in patients with giant hepatic hemangiomas.

## 5. Conclusions

The findings of this study suggest that transarterial bleomycin–lipiodol embolization is a safe and feasible procedure for the treatment of giant hepatic hemangiomas. The low incidence of serious complications and the high degree of drug coverage observed indicate the potential effectiveness of this approach. However, due to the retrospective nature of the study and the limited long-term follow-up, further prospective studies are needed to validate these results and provide a more comprehensive assessment of the safety and efficacy of this treatment option.

## Figures and Tables

**Figure 1 medicina-59-01358-f001:**
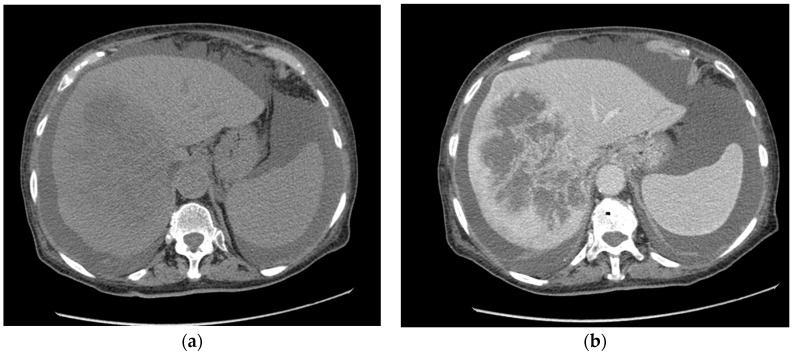
Giant hepatic hemangioma in the right liver lobe—abdomen CT scan. (**a**)—pre-contrast CT, (**b**)—post-contrast CT.

**Figure 2 medicina-59-01358-f002:**
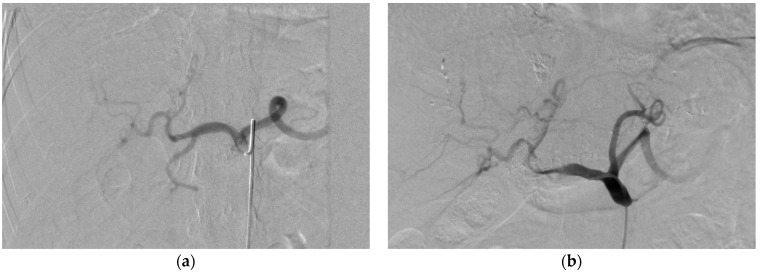
Illustration of common hepatic artery dissection. (**a**)—common hepatic artery before embolization, (**b**)—common hepatic artery after embolization.

**Figure 3 medicina-59-01358-f003:**
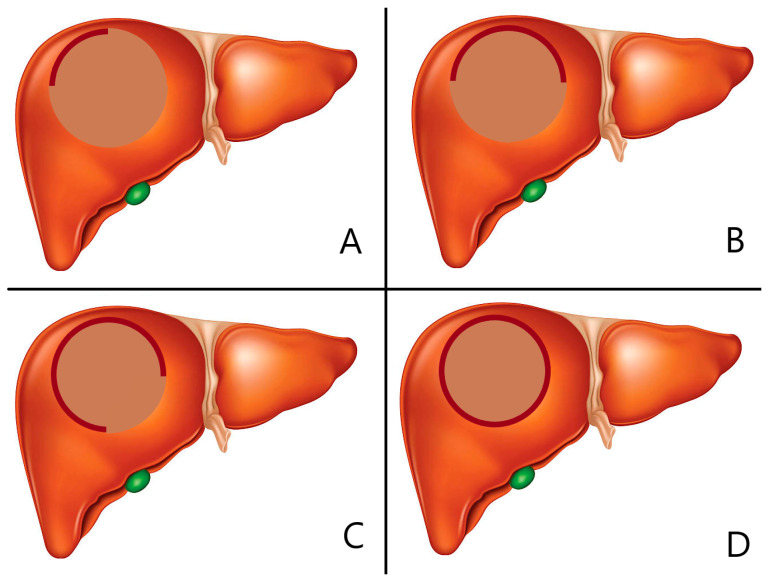
Illustration of drug coverage grade according to hemangioma border coverage. (**A**): Grade 1. less than 25%, (**B**): Grade 2. 25–50%, (**C**): Grade 3. 50–75%, (**D**): Grade 4. more than 75%.

**Figure 4 medicina-59-01358-f004:**
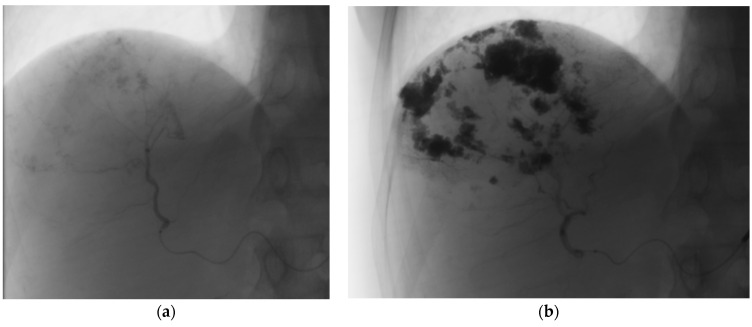
Illustration of complete drug coverage—Grade 4. (**a**)—hemangioma before embolization, (**b**)—hemangioma after embolization.

**Table 1 medicina-59-01358-t001:** Drug coverage grade illustrates the pattern of bleomycin–lipiodol distribution according to hemangioma border coverage. Grade 1: less than 25% of the rim. Grade 2: 25–50% of the rim. Grade 3: less than 75% of the rim. Grade 4: complete rim [12]. (N = 129).

Grade	Frequency	Percent
1	7	5.4
2	10	7.8
3	12	9.3
4	100	77.5

**Table 2 medicina-59-01358-t002:** The severity of the pain in post-embolization syndrome. Grade 0: no pain. Grade 1: mild pain requiring administration of acetaminophen (Paracetamol). Grade 2: moderate pain requiring administration of ketoprofen. Grade 3: severe pain requiring administration of opioids. (N = 124).

Grade	Frequency	Percent
0	70	56.45
1	35	28.23
2	11	8.87
3	8	6.45

**Table 3 medicina-59-01358-t003:** Frequency and mean ± standard deviation (SD) of patient’s basic characteristics.

Characteristics	Group	Frequency	Percent
Sex	women	49	67.1
	man	24	32.9
Age	<40 years	5	6.9
	40–49 years	22	30.1
	50–60 years	20	27.4
	>60 years	26	35.6
Days of hospitalization	<3	2	1.6
	3–5	112	86.8
	>5	15	11.6
Drug coverage grade	<25%	7	5.4
	25–50%	10	7.8
	50–75%	12	9.3
	>75%	100	77.5
Lobe	Right	50	68.5
	Left	21	28.8
	Both	2	2.7
Number of procedures	1	33	45.2
	2	27	37
	3	10	13.7
	4	3	4.1

## Data Availability

Data available on request.

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
