# Peer review of "Safety and Feasibility of Transarterial Bleomycin–Lipiodol Embolization in Patients with Giant Hepatic Hemangiomas"

_medicina, 2023, doi:10.3390/medicina59081358_

Round 1

Author Response

Dear Reviewer,

We would like to express our sincere appreciation for your insightful review of the article titled "Safety and Feasibility of Transarterial Bleomycin–Lipiodol Embolization in Patients with Giant Hepatic Hemangiomas." Your feedback has been immensely valuable, and we highly value the time and effort you dedicated to evaluating our work. We have carefully considered your comments and suggestions, and we would like to respond to them as follows:

1. We have thoroughly reviewed every chemoembolization procedure conducted at our center since 2014 and included patients in this study based on the specified criteria. These criteria involved patients who underwent chemoembolization for giant hepatic hemangiomas at our center between 2014 and 2023, and patients with confirmed diagnosis of hepatic hemangiomas through imaging techniques, including ultrasound, computer tomography (CT), or magnetic resonance imaging (MRI). The exclusion criterion for this study was the presence of atypical angiographic appearance of the hemangioma in the arteriography. We believe that adhering to these criteria has helped ensure the relevance and validity of our study findings.

2. The new insights gathered in our study conclude that chemoembolization of giant liver hemangiomas is a safe and effective method of treatment. These additional references and insights have been duly incorporated into the conclusion paragraph to provide a more comprehensive overview of the findings.

3. We have revised the introduction of the paper to provide more comprehensive details about giant hepatic hemangioma, Bleomycin, Lipiodol, and their respective mechanisms of action. This addition will enhance the background knowledge and understanding of the topic for readers.

4. Thank you for your suggestion regarding the inclusion of a separate table for the comprehensive set of blood tests, liver function tests, and coagulation tests. We appreciate your feedback, and while these tests were conducted as part of the study, their primary aim was to identify any contraindications to the procedure rather than being directly relevant to the specific condition under investigation. Therefore, in order to maintain focus on the key findings and outcomes related to the efficacy and safety of the chemoembolization procedure for giant hepatic hemangiomas, we have decided not to include a separate table. We are committed to providing a clear and concise presentation of the research.

5. In our study, the method used to determine pain severity was the Verbal Rating Scale (VRS). The VRS is a commonly used subjective pain assessment tool where patients are asked to verbally rate their pain on a scale, typically ranging from "no pain" to "worst imaginable pain." The VRS allows individuals to express their pain intensity using descriptive terms, providing valuable information for pain assessment and management.

6. Please note that the information provided in the results paragraph constitutes the statistical section of the study. The focus of this section is on presenting descriptive statistics and distribution patterns rather than exploring correlations or establishing causal relationships. The primary goal of our study was to provide descriptive insights into the safety and feasibility of the chemoembolization procedure. However, if you have any specific questions or requests regarding the statistical analysis or any other aspects you would like to explore, please let us know, and we will be glad to assist you further.

7. We have included a dedicated paragraph in the conclusions section that addresses how to overcome severe complications associated with the Bleomycin-Lipiodol mixture. This paragraph provides insights into the necessary measures to mitigate complications and enhance the safety of the procedure.

Once again, we extend our sincere gratitude for your valuable review. Your expertise and attention to detail have undoubtedly strengthened the quality and clarity of our manuscript. We remain committed to addressing any remaining concerns and ensuring that the final version of the article meets the highest standards of scientific rigor.

Reviewer 2 Report

Thank you for the opportunity to review this interesting paper. 

The authors present a nice and scientifically sound retrospective evaluation of their experience with TACE in giant hepatic hemangiomas. 

However, there is existing evidence of this procedure from other studies including at least one bigger multicenter-study, that needs to be cited and discussed (PMID 27352340, 23580121, 36013000).

As presented, the study lacks novelty. 

The manuscrips also lacks information on clinical presentation of the patients. 

The images provided are nice but I would suggest more images.

Furthermore, with a small number of only n=73 patients I suggest rounding percentages up. 

In the introduction it says "Transarterial Chemoembolization (TACE) was performed using 7-20 cc of Lipiodol mixed with 1500 IU of Bleomycin" and the average Bleomycin dose per procedure is described to be 13015.625 IU, please explain your exact administration regimen. 

English language is fine

Author Response

Dear Reviewer,

Thank you for your insightful review of the article titled "Safety and Feasibility of Transarterial Bleomycin–Lipiodol Embolization in Patients with Giant Hepatic Hemangiomas." We appreciate the time and effort you dedicated to evaluating our work and providing valuable feedback. We would like to respond to your specific points as follows:

  1. We have taken your suggestion into account, and the information regarding all patients with giant hemangioma experiencing symptoms such as abdominal pain, nausea, abdominal distension, dyspepsia, or early satiety prior to TACE has been included in the results section. This addition provides a comprehensive understanding of the clinical presentation of the patients.
  2. Following your recommendation, we have included an additional table containing images of giant hepatic hemangioma in CT. This addition enhances the visual representation and aids in better illustrating the characteristics of the lesions under investigation.
  3. We have reviewed the percentages as per your suggestion, and they have been appropriately adjusted to reflect the data accurately. Rounding up the percentages provides a clearer representation of the distribution patterns observed in the study.
  4. Regarding the administration regimen of Bleomycin, we have added an important note emphasizing that the specific dosage and administration were operator-dependent. The discretion of the operator played a crucial role in determining the need for additional drug injection based on the sufficiency of the hemangioma coverage achieved. This individualized approach allowed for tailored treatment, ensuring the best possible outcome for each patient.

Once again, we sincerely appreciate your thorough review and constructive feedback. Your valuable contributions have greatly improved the quality of our manuscript. We remain committed to addressing any remaining concerns and ensuring that the final version of the article meets the highest standards of scientific rigor.

Round 2

Reviewer 1 Report

Thanks, authors for their response on the required modifications.